# THRUST: ADAPTIVELY PROPELS LARGE LANGUAGE MODELS WITH EXTERNAL KNOWLEDGE

## ABSTRACT

Large-scale pre-trained language models (PTLM) have achieved great success in various natural language processing (NLP) tasks. Much evidence shows that PTLMs already encode rich knowledge themselves, but knowledge stored in PTLMs can be opaque and static, making external knowledge retrieval necessary. However, there are two major challenges when using external knowledge. First, knowledge indexing and retrieving on large-scale knowledge bases are time costly. Second, knowledge retrieved could be noisy and sometimes misleading. Motivated by the observation that external knowledge is not always required by PTLMs, we investigate an effective and efficient way to apply knowledge only when the knowledge is essential. Specifically, we propose instance-level adaptive propulsion of external knowledge (IAPEK), where we score each instance on whether the PTLMs need the support of external knowledge. To achieve this goal, we design a novel metric, ***Thrust***, which leverages the distribution estimation on seen/training instances. Extensive experiments demonstrate that we can achieve significantly higher cost-efficiency through ***Thrust*** compared to the naive usage of external knowledge on 88% of the evaluated tasks with 26% average performance improvement. Such findings further shed light on the real-world practice of knowledge-enhanced LMs with a limited budget for knowledge seeking due to computation latency or costs [1].

## 1 INTRODUCTION

Knowledge plays an important role in solving natural language processing (NLP) tasks, where encyclopedic or commonsense knowledge is commonly required to answer questions from various tasks (Yin et al., 2022). In recent years, the emergent advance of pre-trained language models (PTLM) has demonstrated great improvement on various tasks (Devlin et al., 2019; Radford et al., 2019; Liu et al., 2019; Raffel et al., 2020; Brown et al., 2020). Evidence also show that PTLMs contain rich encyclopedic (Petroni et al., 2019) or commonsense (Kocijan et al., 2019) knowledge themselves. However, such implicit knowledge embedded in the model's hidden states can be opaque, static, and inefficient to utilize (Khattab et al., 2022). These issues motivate the common practice on seeking external knowledge (Xu et al., 2021; Verga et al., 2021; Paranjape et al., 2022) in NLP. A typical line of work focuses on retrieval-based methods, where knowledge is retrieved by a stand-alone retriever from external knowledge bases and then used to augment the inference models (i.e., Reader) such as PTLMs (Karpukhin et al., 2020; Gao & Callan, 2021; Khattab & Zaharia, 2020).

However, there are several limitations with the usage of external knowledge: (i) performance on the downstream tasks is not commonly revealed. Metrics of the common benchmarks (e.g., MS-MARCO (Nguyen et al., 2016), BEIR (Thakur et al., 2021)) measure the quality of retrieval (e.g., Recall@50, nDCG@10). Although retrieving the relevant content may positively relate to the downstream performance, not reporting the downstream performance, especially for the out-of-domain tasks, limits the exploration of how to utilize the external knowledge in practice; (ii) the external knowledge can be noisy or unnecessary. On the retriever side, though concurrent retrievers achieve great performance on various tasks, the noise can still exist. For instance, ColBERT v2 (Santhanam et al., 2022) achieved 68.9 Success@5 on Natural Question (Kwiatkowski et al., 2019), which suggests that gold documents do not appear in the top 5 retrieved documents for 31.1% of the queries.

---

[1]The code and data will be released upon acceptance.

- **Query:** Who is the first woman to get the Nobel prize in physics?
- **Answer:** Marie Curie
- **Direct Prediction:** Marie Curie

- **Knowledge #1**: Wilhelm Conrad Rontgen discovered X-ray ... which made him the first winner of the Nobel prize in physics in 1901 ...
- **Prediction:** Wilhelm Conrad Rontgen

- **Knowledge #2**: Marie Curie was the first woman to be awarded a Nobel prize. Curie and her husband declined to go to Stockholm ...
- **Prediction**: Marie Curie

Figure 1: The predictions from OPT (175B version) with/without external knowledge retrieved by DPR (Karpukhin et al., 2020) from Wikipedia paragraphs. Although the top retrieved paragraphs are relevant, since the internal knowledge is already sufficient, the external knowledge can either be misleading (potentially due to the effect of *misprime* (Kassner & Schütze, 2020)) or less useful.

Considering the limited input token length, the most useful documents may not be included during prediction. Not to mention that others can be noise to the model. On the model side, PTLMs with growing capacity, from millions (e.g., BERT (Devlin et al., 2019)) to billions of parameters (e.g., OPT (Zhang et al., 2022b)), may solve the queries directly without external knowledge, which makes it unnecessary to seek external knowledge and signifies the noise issue. The instance shown in Figure 1 demonstrates the noise and inefficiency issues. OPT (the version with 175 billion parameters) can directly give the correct answer without any external knowledge. However, with top external knowledge retrieved by DPR (Karpukhin et al., 2020) from Wikipedia paragraphs, the external knowledge can be useless or even lead to wrong predictions.

Intuitively, a solution to the noise and inefficiency issues is only seeking external knowledge when it is necessary. In this work, we capture this intuition by proposing **I**nstance-level **A**daptive **P**ropulsion of **E**xternal **K**nowledge (**IAPEK**) to reduce the effect of noise in external knowledge and improve the cost-efficiency of knowledge augmentation. In detail, for each instance of a given task, we compute a confidence score measuring how likely it can be solved directly with respect to a given model and reject the use of external knowledge when the score is high. We design a simple and lightweight metric ***Thrust*** to serve such a purpose by leveraging the estimation of the instance distribution in the eyes of the target models. To comprehensively understand the effectiveness of ***Thrust***, we first create a large-scale benchmark examining the downstream performance of the task-plus-knowledge paradigm with (i) tasks with different formats and types (e.g., multiple-choice classification (MC classification) and open-domain question answering (open-domain QA)); (ii) knowledge with different formats and from different resources (e.g., knowledge graphs, Wikipedia paragraphs, and human annotations). Next, with models that can utilize external knowledge, we evaluate the effectiveness of ***Thrust*** by showing that it can boost the performance of various tasks under various settings, such as injecting external knowledge to different portions of the test instances.

Extensive experiments show that ***Thrust*** can improve the cost-efficiency of seeking and using external knowledge on 88% cases with 26% average performance improvement through identifying the instances that mostly require knowledge. We can also observe that, with ***Thrust***, we can achieve higher performance than injecting external knowledge for all the instances, where models are benefited from both the performance and efficiency aspects.

## 2 OUR METHOD

### 2.1 INSTANCE-LEVEL ADAPTIVE PROPULSION OF KNOWLEDGE

We first define **IAPEK** as follows: for each query $q_i$ in a given test set $\mathcal{D} = \{q^{(1)}, q^{(2)}, \ldots\}$, let $f(q)$ denotes the scoring function of the necessity of external knowledge, we extract the corresponding scores $\mathcal{S} = \{f(q)^{(1)}, f(q)^{(2)}, \ldots\}$. With $\mathcal{S}$, we re-rank the test set into $\mathcal{D}' = \{q'^{(1)}, q'^{(2)}, \ldots\}$. Given any threshold $t \in R$, we sample a subset $\mathcal{D}_k = \{q_k^{(1)}, q_k^{(2)}, \ldots\}$ as that with highest knowledge need, where for each $q_k \in D_k$, $f(q_k) > t$. Empirically, we can set $t$ as a particular percentile of $\mathcal{S}$, e.g., top 25% of $\mathcal{S}$. Next, for each instance in $\mathcal{D}_k$, we seek for external knowledge pieces and

Figure 2: Figurative illustration of the intuition behind proposing **Thrust**. We represent the instance query vectors as triangles and instance clusters with different labels as circles with different colors. In the *controversial* and *no knowledge* cases, the knowledge contained internally is less likely to answer the query successfully. In contrast, if the model finds the query similar to various seen instances, the internal knowledge is likely to exist.

augment each query $q_k$ to $q_{k+}$. We combine updated $\mathcal{D}_{k+}$ and original unsampled instances $\mathcal{D} \setminus \mathcal{D}_k$ to the new knowledge augmented dataset $\mathcal{D}_+$ to apply to inference models.

## 2.2 THRUST

We then introduce a novel empirical method, **Thrust**, to perform the proposed instance-level adaptive propulsion of external knowledge (**IAPEK**). We design **Thrust** to measure how likely the given query can be solved by the internal knowledge of the target model. Intuitively, there are two cases where models can fail to answer a query with internal knowledge: (i) the model has no relevant knowledge and is not familiar with the query semantics or inference types; (ii) the model faces controversial knowledge, where the query may have similar semantics with different kinds of seen questions that potentially require different reasoning to solve.

As shown in Figure 2, the degree of difference of queries and clusters of seen instances can measure how well the internal knowledge of the model covers such queries. Motivated by the intuition, the **Thrust** score is computed as follows: given a task $\mathcal{T}$, we first group the instances by their labels. For QA tasks, we regard all instances as having a single dummy label. Then, we embed instances within each group with an embedding function $f(.)$. Next, we form groups for each task $\mathcal{T}$ as $g_l = \{(f(x_i), y_i)) \mid y_i = l\}$. With the groups in hand, we perform k-means clustering to proceed each group into K clusters and instances of task $T$ are represented by the clusters $\left\{ C^j(g_l) \right\}_{j=1}^{K}$ for $l = 1, 2, ..., N$, with $m^j(g_l)$ as the corresponding centroids and $\vec{m}^j(g_l)$ as the vectorized centroids pointing from $\vec{0}$ to the centroids[2].

For some test time input $x \in \mathcal{T}$, where the label is unknown, let $\vec{f}(x)$ denote the vectorized features pointing from $\vec{0}$ to the embedded $f(x)$, the directed distance between the instance and the clusters are then defined as

$$\vec{d}(f, x, j, l, \mathcal{T}) \triangleq \vec{m}^j(g_l) - \vec{f}(x). \tag{1}$$

Influence of each cluster towards the query is considered as a vector with the same direction of $\vec{d}$ and re-weighted by the size of the cluster over the square Euclidean distance. **Thrust** score for an input $x$ is then computed as

$$s_{Thrust} \triangleq \frac{1}{N \cdot K} \| \sum_{l=1}^{N} \sum_{j=1}^{K} \frac{|C^j(g_l)| \cdot \vec{d}(f, x, j, l, \mathcal{T})}{\|\vec{d}(f, x, j, l, \mathcal{T})\|^3} \|. \tag{2}$$

[2]We use the last layer of hidden states as the embedding function. For T5-based models, we use the last layers of the decoders. We empirically set the number of clusters as the $max(\sqrt[4]{|D_{\mathcal{T}}^{tr}|}, 3)$, where $D^{tr}$ denotes the training set of task $\mathcal{T}$.

| Dataset | Type | Source | Train # | Test # | Query Len | Know. Len | Ans. Len |
|---------|------|--------|---------|--------|-----------|-----------|----------|
| AGNews | MC | gold | 120,000 | 7,600 | 8.1 | 35.9 | 1.0 |
| e-SNLI | NLI | human | 259,999 | 9,824 | 24.9 | 14.3 | 1.0 |
| StrategyQA | Binary | human | 2,290 | 229 | 10.8 | 33.5 | 1.0 |
| CIKQA | Binary | KG | 4,818 | 604 | 18.2 | 28.0 | 1.0 |
| BoolQ | Binary | retriever | 9,437 | 3,270 | 9.8 | 113.8 | 1.0 |
| ARC-E | MC | retriever | 2,251 | 570 | 23.1 | 238.2 | 4.2 |
| ARC-C | MC | retriever | 1,119 | 299 | 26.2 | 240.5 | 5.5 |
| HotpotQA | QA | gold | 90,447 | 7,405 | 19.0 | 56.3 | 2.5 |
| NQ | QA | retriever | 96,676 | 6,468 | 10.1 | 588.9 | 2.3 |
| Web Questions | QA | retriever | 2,474 | 278 | 7.8 | 117.3 | 4.3 |
| Curated TREC | QA | retriever | 1,125 | 116 | 8.4 | 116.5 | 7.7 |
| TriviaQA | QA | retriever | 78,785 | 6,760 | 15.0 | 117.6 | 27.5 |

Table 1: Statistics of the selected datasets. ARC-E & ARC-C denote the easy and hard ARC datasets as defined in the original work. MC, NLI, Binary, QA denote the task types of multiple-choice classification, natural language inference, binary classification, and question answering, respectively. Query/Know./Ans. Len denote the average numbers of words for the queries/knowledge/answers.

## 3 EXPERIMENT

### 3.1 SETUP

We evaluate T5 (Raffel et al., 2020), GPT-J (Wang & Komatsuzaki, 2021), OPT (Zhang et al., 2022b), and UnifiedQA (Khashabi et al., 2020). We test both the zero-shot learning setting (models take the prompt with or without knowledge directly) and the transfer-learning setting (models are fine-tuned with instances containing external knowledge) to examine the proper way to use external knowledge. For the zero-shot setting, we test on T5, GPT-J, and OPT (30 billion parameter version). For the transfer-learning setting, we test on UnifiedQA with different scales.

After that, we evaluate the effectiveness of *Thrust*, where we try to simulate the real-world case where we have limited bandwidth or budget to retrieve external knowledge. Specifically, we are only allowed to conduct the knowledge retrieval for 25%, 50%, and 75% of instances. As introduced in Section 2.1, we use *Thrust* to rank the instances by their need for external knowledge and select the instances from the most necessary ones. The threshold can be considered as expecting 100-X% savings ($X = 25, 50, 75$) in continuously incoming future test inputs. We compare the performance difference between *Thrust* and random sampling[3].

For MC classification tasks, we follow previous works to use accuracy as the evaluation metric. For open-domain QA, we report the QA-F1 scores of the models under different settings. The QA-F1 score for question answering measures the max uni-gram overlap between the model prediction and all gold answer candidates. Denoting the gold answer set as $\mathcal{G}$ and uni-gram tokens of each query and each corresponding gold answer as $\mathcal{T}_q$ and $\mathcal{T}_g$, the QA-F1 score can then be written as QA-F1 $= \max_{g \in \mathcal{G}}(\mathcal{T}_q \cap \mathcal{T}_g)/(\mathcal{T}_q \cup \mathcal{T}_g)$. We test on two settings: *without knowledge* and *with knowledge*. For the former one, we directly pass the prompt-decorated queries to the model to retrieve the choices with the highest probability or the answer to the questions. For the latter one, we add the prompt-decorated queries, one piece of knowledge, and *Answer:* into three lines and pass them all to the models.

### 3.2 DATASETS

We first prepare a benchmark containing knowledge-intensive tasks from two main types: MC classification and open-domain question answering, with seven and five tasks, respectively. We unify the instances into the same format, where each contains: (i) a query: a piece of text containing a question or the sentences to be classified; (ii) an answer: either the label words or the answers to the questions in the query; (iii) knowledge: one piece of potentially helpful knowledge for the query, which is either inherently relevant due to the task design, annotated by humans, or retrieved from Wikipedia paragraphs with DPR. Details of the selected datasets and definitions of external knowledge for each task are as follows:

---

[3]We also involve BM25 as an alternative to perform **IAPEK**. The comparison can be found in Appendix.

**Multiple-choice classification.** For MC classification, each query $q$ includes a sentence or a question and requires models to extract one correct answer from a set of candidate answers (e.g., *yes* or *no*). The selected tasks are (i) AGNews (Zhang et al., 2015): AGNews is a classic text classification task in NLP, where models are required to classify if a piece of news belongs to which among *political, sports, business,* or *technology* categories. We regard the titles of the news as the queries since they may already contain sufficient information for classification (e.g., *Olympic history for India* belongs to *Sports*); Then, the content of the news is considered as the gold external knowledge extracted from the task design; (ii) e-SNLI (Camburu et al., 2018): e-SNLI is a natural language inference (NLI) task exploiting the role of explanations for the task of textual entailment. We concatenate the original two sentences with a blank and add a question about if entailment exists to form the query. Naturally, the human-providing explanations are considered a strong source of external knowledge; (iii) StrategyQA (Geva et al., 2021): StrategyQA is a challenging multi-hop reasoning dataset that requires models to answer creative questions (e.g., *Did Aristotle use a laptop?*) through strategical inference from implicit reasoning steps. We regard the original creative questions as queries and human-written explicit facts (e.g., *Aristotle was born in ...*) as external knowledge, which is again powerful and expected to contain little noise; (iv) CIKQA (Zhang et al., 2022a): CIKQA is a commonsense inference task in the format of multiple-choice question answering, combining the tasks of pronoun coreference resolution, commonsense QA (Talmor et al., 2019), COPA (Roemmele et al., 2011), and questions mined from ATOMIC knowledge graph (Sap et al., 2018). We regard the original questions as queries and the supporting commonsense knowledge extracted from knowledge graphs (KGs) in the original work as the external knowledge; (v) BoolQ (Clark et al., 2019): BoolQ is a question that contains encyclopedic questions that require models to answer *yes* or *no*. Following (Khashabi et al., 2020), we use the Wikipedia paragraphs retrieved by DPR as the external knowledge, which can be potentially noisy; (vi) ARC-E & ARC-C (Clark et al., 2018): ARC is a challenging multiple-choice question answering dataset that requires powerful knowledge understanding and reasoning, which is partitioned to an Easy set and a Challenge set (denoted as ARC-E & ARC-C, respectively), where the Challenge set questions are answered incorrectly by the retrieval-based or co-occurrence-based algorithms tested by the original authors. Similarly, we use the Wikipedia paragraphs retrieved by DPR as external knowledge.

**Open-domain QA.** For open-domain QA, each query $q$ contains an open question that typically requires solving an encyclopedic or commonsense inference. The generated answers can either be a few phrases or a single sentence. The involved datasets are HotpotQA (Yang et al., 2018), Natural Questions (NQ) (Kwiatkowski et al., 2019), Web Questions (Berant et al., 2013), Curated TREC (Baudiš, 2015), and TriviaQA (Joshi et al., 2017). We use Wikipedia paragraphs retrieved by DPR as the external knowledge as a common practice (Yin et al., 2022), except for HotpotQA, where we use the passages the queries are generated from as a gold knowledge resource.

The statistics of the involved datasets are reported in Table 1. We collect a benchmark with various datasets of different types, formats, and knowledge sources, where we will then evaluate the effectiveness of **IAPEK**.

## 3.3 Using External Knowledge

Table 2 presents the model performance on both the MC classification and open-domain QA tasks. For the MC classification tasks, we can observe that: (i) for the zero-shot setting, models do not consistently get benefit from external knowledge. In addition, the ability to utilize external knowledge is also not clearly improved as the parameter size grows, which indicates that simply using larger models may not be the solution for better using the knowledge; (ii) for the transfer-learning setting, although *AGNews*, *e-SNLI*, *CIKQA*, and *StrategyQA* are not seen during the training of UnifiedQA models, we can observe that models achieve better performance than vanilla T5 models at different sizes. Under the *with knowledge* case, the UnifiedQA models achieve significant improvement for utilizing external knowledge compared to the zero-shot models, *UnifiedQA-3b* achieves the best performance on all the tasks, which indicates that models can learn and transfer the ability to utilize external knowledge with instances containing external knowledge.

For open-domain QA datasets, we can observe that: (i) similarly, models fail to get benefit from external knowledge in 11 out of 25 cases under the zero-shot setting. Surprisingly, the smallest model *T5-base* gets benefits for all the tasks, but the largest model (OPT-30b) gets worse performance with knowledge for all tasks. The reason behind this can be that, since T5-base does not

| Model | parameters | AGNews | e-SNLI | CIKQA | StrategyQA | BoolQ | ARC-E | ARC-C |
|---|---|---|---|---|---|---|---|---|
| **Zero-shot** | | | | | | | | |
| T5-base | 220M | 30.2 \| 44.4 | 65.2 \| 65.1 | 51.5 \| 51.8 | 54.1 \| 50.2 | 48.3 \| 38.9 | 27.8 \| 28.8 | 31.4 \| 29.4 |
| T5-large | 770M | 25.8 \| 25.2 | 65.7 \| 65.7 | 50.0 \| 50.0 | 53.3 \| 53.3 | 37.8 \| 38.6 | 25.1 \| 27.7 | 27.7 \| 24.7 |
| T5-3b | 3B | 27.9 \| 39.1 | 57.6 \| 61.5 | 52.6 \| 50.5 | 44.5 \| 48.9 | 56.6 \| 45.3 | 25.8 \| 26.0 | 26.4 \| 28.4 |
| GPT-J | 6B | 25.1 \| 26.9 | 40.8 \| 37.0 | 49.8 \| 50.7 | 47.2 \| 55.9 | 60.2 \| 47.2 | 25.4 \| 29.5 | 28.4 \| 27.1 |
| OPT-30b | 30B | 25.0 \| 25.0 | 65.7 \| 65.7 | 50.0 \| 50.0 | 53.3 \| 53.3 | 37.8 \| 37.8 | 27.4 \| 27.7 | 25.8 \| 26.4 |
| **Transfer-learning** | | | | | | | | |
| UnifiedQA-base | 220M | 46.6 \| 35.7 | 38.5 \| 70.2 | 56.0 \| 59.6 | 48.5 \| 57.2 | 60.4 \| 80.8 | 50.2 \| 61.6 | 44.8 \| 45.2 |
| UnifiedQA-large | 770M | 71.0 \| 67.9 | 42.8 \| 74.2 | 59.6 \| 62.1 | 48.5 \| 66.4 | 59.8 \| 84.5 | 64.0 \| 66.0 | 55.2 \| 49.5 |
| UnifiedQA-3b | 3B | 75.7 \| 84.5 | 62.2 \| 89.6 | 61.3 \| 66.9 | 57.6 \| 83.4 | 61.5 \| 87.8 | 73.7 \| 76.5 | 64.5 \| 64.2 |

| Model | parameters | Web Questions | Curated TREC | HotpotQA | NQ | TriviaQA |
|---|---|---|---|---|---|---|
| **Zero-shot** | | | | | | |
| T5-base | 220M | 6.7 \| 8.7 | 2.5 \| 3.8 | 6.0 \| 9.1 | 1.9 \| 6.0 | 8.9 \| 13.1 |
| T5-large | 770M | 5.7 \| 7.4 | 1.9 \| 3.0 | 5.1 \| 6.6 | 1.6 \| 2.7 | 8.3 \| 9.0 |
| T5-3b | 3B | 4.9 \| 4.0 | 2.0 \| 1.0 | 4.9 \| 6.8 | 1.7 \| 6.6 | 8.3 \| 5.6 |
| GPT-J | 6B | 4.3 \| 6.3 | 7.4 \| 2.7 | 5.9 \| 5.1 | 10.9 \| 6.9 | 1.7 \| 2.1 |
| OPT-30b | 30B | 18.3 \| 6.3 | 16.0 \| 2.4 | 11.4 \| 2.4 | 5.3 \| 2.1 | 16.3 \| 6.9 |
| **Transfer-learning** | | | | | | |
| UnifiedQA-base | 220M | 10.6 \| 44.2 | 3.6 \| 36.9 | 13.1 \| 40.3 | 3.0 \| 34.6 | 11.7 \| 65.6 |
| UnifiedQA-large | 770M | 12.9 \| 46.5 | 8.2 \| 36.6 | 14.2 \| 42.7 | 3.8 \| 36.3 | 13.7 \| 74.6 |
| UnifiedQA-3b | 3B | 11.5 \| 48.1 | 9.9 \| 41.8 | 17.0 \| 47.0 | 4.4 \| 37.6 | 18.6 \| 80.0 |

Table 2: Performance of various models on the MC classification tasks (accuracy) and open-domain QA tasks (QA-F1). Performances without/with knowledge external knowledge are presented before/after the vertical bar, respectively. UnifiedQA-X denotes T5 models with corresponding sizes fine-tuned on the UnifiedQA dataset.

| Dataset | UnifiedQA-base | | | UnifiedQA-large | | | UnifiedQA-3b | | |
|---|---|---|---|---|---|---|---|---|---|
| | 25% | 50% | 75% | 25% | 50% | 75% | 25% | 50% | 75% |
| AGNews | 50.7 \| 55.6 | 52.8 \| 56.3 | 55.0 \| 56.8 | 70.2 \| 69.1 | 69.4 \| 70.2 | 68.7 \| 70.6 | 77.9 \| 78.4 | 80.1 \| 80.4 | 82.3 \| 82.3 |
| e-SNLI | 46.5 \| 66.6 | 54.4 \| 68.3 | 62.3 \| 69.6 | 50.7 \| 71.1 | 58.5 \| 72.2 | 66.4 \| 73.2 | 69.1 \| 86.3 | 75.9 \| 87.5 | 82.8 \| 88.8 |
| CIKQA | 56.9 \| 59.6 | 57.8 \| 59.6 | 58.7 \| 59.9 | 60.2 \| 62.1 | 60.8 \| 62.3 | 61.5 \| 62.4 | 62.7 \| 66.9 | 64.1 \| 66.9 | 65.5 \| 66.9 |
| StrategyQA | 50.7 \| 55.6 | 52.8 \| 56.3 | 55.0 \| 56.8 | 52.9 \| 62.1 | 57.4 \| 65.3 | 61.9 \| 65.9 | 64.1 \| 74.3 | 70.5 \| 81.4 | 77.0 \| 82.9 |
| BoolQ | 65.5 \| 76.2 | 70.7 \| 79.9 | 75.8 \| 80.9 | 65.9 \| 77.7 | 72.1 \| 81.3 | 78.3 \| 84.4 | 68.1 \| 79.1 | 74.6 \| 85.7 | 81.2 \| 87.1 |
| ARC-E | 50.7 \| 55.6 | 52.8 \| 56.3 | 55.0 \| 56.8 | 64.5 \| 64.6 | 65.0 \| 64.7 | 65.5 \| 65.1 | 74.4 \| 74.6 | 75.1 \| 74.9 | 75.8 \| 75.1 |
| ARC-C | 44.9 \| 43.8 | 45.0 \| 44.5 | 45.1 \| 44.8 | 53.8 \| 50.8 | 52.3 \| 51.2 | 50.9 \| 51.5 | 64.5 \| 63.9 | 64.4 \| 64.9 | 64.3 \| 65.6 |
| WQ | 19.2 \| 26.3 | 27.5 \| 42.1 | 35.8 \| 43.8 | 22.5 \| 38.5 | 30.5 \| 39.0 | 38.5 \| 46.0 | 20.9 \| 19.3 | 30.0 \| 35.4 | 39.1 \| 46.4 |
| TREC | 13.5 \| 33.6 | 21.3 \| 36.4 | 29.1 \| 36.9 | 30.8 \| 32.7 | 32.7 \| 36.0 | 34.6 \| 36.3 | 19.6 \| 37.8 | 27.0 \| 40.6 | 34.4 \| 40.9 |
| HotpotQA | 25.2 \| 32.9 | 30.2 \| 35.5 | 35.2 \| 37.8 | 26.7 \| 35.2 | 32.1 \| 37.5 | 37.4 \| 40.2 | 24.9 \| 41.9 | 32.3 \| 43.9 | 39.7 \| 45.7 |
| TriviaQA | 32.0 \| 52.7 | 43.2 \| 56.4 | 54.4 \| 60.0 | 32.4 \| 59.7 | 46.4 \| 64.3 | 60.5 \| 71.8 | 39.2 \| 68.3 | 52.8 \| 71.0 | 66.4 \| 73.4 |
| NQ | 20.0 \| 33.0 | 24.9 \| 33.5 | 29.7 \| 33.9 | 12.0 \| 34.8 | 20.1 \| 35.2 | 28.2 \| 35.7 | 12.8 \| 35.9 | 21.1 \| 36.5 | 29.4 \| 37.0 |

Table 3: Performance of **IAPEK** leveraging *Thrust*. With 25%, 50%, and 75% percent instances augmented with their corresponding knowledge, performances of random/***Thrust*** are presented before/after the vertical bar. If performance increases with ***Thrust***, the score will be marked in green and otherwise in red. WQ, TREC denote the tasks of Web Questions, Curated TREC, respectively.

have enough internal knowledge, any relevant external knowledge can help. As a comparison, the OPT-30b model already contains rich knowledge, and thus the external knowledge may only be introducing extra noise if the model does not learn to utilize knowledge; (ii) under the transfer-learning setting, UnifiedQA-based models get significant benefit from the external knowledge, again showing the effectiveness of helping models to learn to use knowledge.

In conclusion, we find that fine-tuning on instances containing external knowledge is an effective way to help models gain performance increase from external knowledge at test time. Since the pre-condition of using IAPEK is that the model can utilize external knowledge well, we conduct experiments with UnifiedQA only when evaluating the performance of ***Thrust***.

| Model | *Thrust* > Full | | | *Thrust* < Full | | |
|---|---|---|---|---|---|---|
| UnifiedQA-base | BoolQ(r)* ARC-C(r) | CIKQA(r) TriviaQA(r)* | StrategyQA(h) AGNews(g) | e-SNLI(h) TREC(r) | ARC-E(r)* HotpotQA(g)* | WQ(r)* NQ(r) |
| UnifiedQA-3b | BoolQ(r)* HotpotQA(g) | CIKQA(r) | ARC-C(r) | e-SNLI(h) ARC-E(r)* TriviaQA(r)* | StrategyQA(h)* WQ(r) NQ(r)* | AGNews(g)* TREC(r) |

Table 4: Comparison between using ***Thrust*** and the costly full knowledge usage for models with different scales. The knowledge type is noted in bracket, where *g* denotes gold knowledge, *h* denotes human annotated knowledge, and *r* denotes the knowledge retrieved from Wikipedia passages or knowledge graphs. If the performance difference is less than 1% accuracy or QA-F1, we denote the entry with *. WQ, TREC denote the tasks of Web Questions, Curated TREC, respectively. We find that for many cases ***Thrust*** can outperform full knowledge usage with fewer instances augmented with external knowledge.

### 3.4 PERFORMANCE OF THRUST

From the results in Table 3, we can observe that: (i) ***Thrust*** consistently contributes to the performance from the base to the 3B model. Through clustering the instances, we acquire the whole instance distribution in the eyes of the models. Then with distance to the cluster, ***Thrust*** represents how well the model can categorize a new query vector and find its similarity with others on the task. Leveraging such information, ***Thrust*** identifies the *no knowledge* and *controversial knowledge* cases well and puts the knowledge into the most necessary ones; (ii) the gain is higher when the portion of augmented instances is smaller. For instance, for UnifiedQA-3b, the gains from ***Thrust*** with 25% instances augmented with knowledge are 6.1%, 13.56% on MC classification and QA tasks, respectively, while for the 75% case, the gains are 2.8% and 6.8%. Such observation shows that ***Thrust*** is most effective on identifying the most necessary cases. One potential reason is that ***Thrust*** is sensitive to the distance change so the isolated instances (*no knowledge case in Figure 2*) can be easily identified; (iii) we also observe consistent failure case on ARC-C. The reason can be that the queries are designed as open questions, and the answers are usually about plans or ideas, not facts, so that it is hard for the small-size models to extract useful information from the seemingly unrelated Wikipedia document. For instance, a query from ARC-C is: *Juan and LaKeisha roll a few objects down a ramp. They want to see which object rolls the farthest. What should they do so they can repeat their investigation?*. The correct and wrong options are *Record the details of the investigation* and *Choose different objects to roll*. For questions of this style, it is even hard for humans to find a relevant Wikipedia page that can help. The failure case further sheds light on the pre-condition of ***Thrust***: we assume that external knowledge is useful.

As for efficiency, following the definition in Section 2.2, at test time, the computation complexity for ***Thrust*** is $O(NK)$, where $N$ is the number of labels and $K$ is the number of clusters, for a five-way classification task, $N \cdot K \approx 15$. However, if we retrieve knowledge from a external corpus, the computation complexity is $O(M)$, where $M \approx 1,000,000$ for DPR. In this case, expected performance improvement is $O(\alpha M - NK)$, with $\alpha$ as the probability of rejecting the knowledge use for one example, which is 0.25 to 0.75 in our experiment. On the other hand, extracting knowledge from other sources (e.g., human annotations for e-SNLI) can be potentially more time-consuming.

## 4 ANALYSIS

### 4.1 COMPARISON WITH FULL KNOWLEDGE USAGE

We denote simply using external knowledge for all instances as a costly but straightforward way of leveraging external knowledge. Since the big models might be sufficient for certain instances and the external knowledge might introduce extra noise, we hypothesize that, in some cases, ***Thrust*** can help identify instances requiring (or not) knowledge and achieve higher overall performance on the whole dataset compared to seeking and adding knowledge indiscriminately. Table 4 presents the comparison between adaptive and indiscriminate knowledge propulsion. ***Thrust*** here denotes the best performance achieved when less than 90% of instances use external knowledge. We could observe that, for 50% and 30% tasks for UnifiedQA-base and UnifiedQA-3b, respectively, ***Thrust*** achieves better

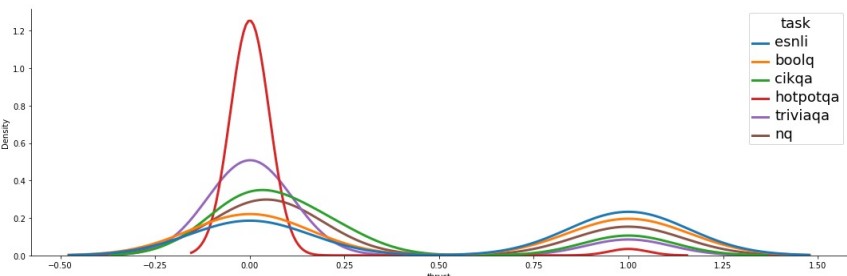

Figure 3: Distribution of **Thrust** scores for various tasks with UnifiedQA-3b to create the instance representation. The distribution is normalized by Kernel Density Estimation. Low scores denote the cases where internal knowledge is not enough, vice versa. We find that **Thrust** scores predict that more cases from HotpotQA require external knowledge and more cases from e-SNLI do not require external knowledge compared to other tasks.

performance than using knowledge for all instances. Such results indicate that **Thrust** can help avoid potential noise. On the other hand, we can also observe that for the *e-SNLI*, *Web Questions*, and *NQ*, the full knowledge setting performs better than **Thrust**. The reason behind this can be that external knowledge is essential and of high quality (e.g., the used knowledge is manually written rather than retrieved). As long as the model can comprehend the external knowledge, seeking and adding more high-quality knowledge always benefit the models.

## 4.2 DISTRIBUTION OF THRUST ACROSS TASKS

To further understand how **Thrust** leads to adaptive knowledge propulsion, we select the top 3 tasks with the highest improvement after applying **IAPEK** with **Thrust** from MC classification and open-domain QA tasks, respectively. The selected tasks are: *e-SNLI*, *BoolQ*, *CIKQA*, *HotpotQA*, *TriviaQA*, and *NQ*. Figure 3 demonstrates the distribution of **Thrust** scores for each of the involved tasks. The scores are cast to $[0, 1]$ with respect to the extremum, and the distribution is normalized by Kernel Density Estimation. From the figure, we can observe that low scores (i.e., the query needs external knowledge) appear commonly for the open-domain QA tasks such as *HotpotQA* and *TriviaQA*. *CIKQA* queries, which are designed to require commonsense knowledge to solve, also need external knowledge for many cases, as predicted by **Thrust**. On the other hand, for *e-SNLI* and *BoolQ*, external knowledge is not always necessary. Such findings demonstrate the potential of using **Thrust** to investigate the characteristics of tasks from the knowledge aspect.

## 4.3 LAYER ABLATION

Since we cast instances into the representation space, a crucial contributing factor for **Thrust** is the layer of the PTLM to use. To investigate the effect of which layer to use, we conduct experiments on UnifiedQA-3b with the same setting as in Section 3.1. Figure 4 presents the performance of adding 25%, 50%, 75% knowledge-augmented instances with **Thrust** using the hidden states of different layers. We can observe that, for most tasks, there is no significant difference across layers, which shows the robustness of **Thrust** and potential capacity to accelerate the computation of **Thrust** of using lower layers. However, for some tasks, such as *StrategyQA* and *Web Questions*, the middle-layer representation may worsen the overall performance. The reason can be that: early layers in the model contain rich semantic information, and later layers contain task-specific information (Lovering et al., 2021), so that both can act as good representation of the instances. However, in the middle layers, rich semantic features are abandoned during extracting task-specific features and task-specific features are also not fully extracted and expressed yet.

## 5 RELATED WORK

**PTLM with external knowledge.** The paradigm of retrieving knowledge from knowledge bases, augmenting PTLMs, and solving downstream tasks has been widely explored in the community of NLP (Lewis et al., 2020; Borgeaud et al., 2022; Izacard et al., 2022). The knowledge bases can

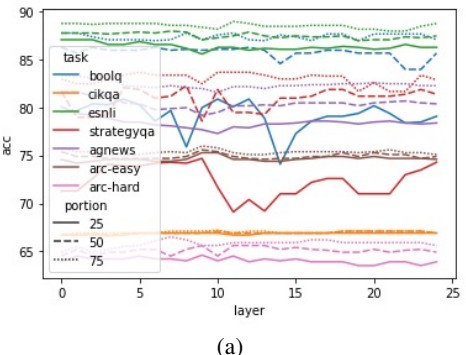 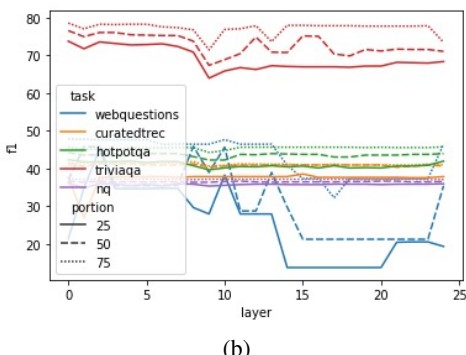

(a)                                          (b)

Figure 4: Layer-wise ablation across tasks and portions of instances augmented with knowledge for (a) MC classification tasks and (b) open-domain QA tasks. The X-axis denotes the layer of Unified-3b decoder used to cast instances into the representation space. We find that for most tasks, the layer number will not be a significant factor. For some tasks (e.g., StrategyQA), choosing middle layers for representation slightly worsens the performance.

range from knowledge graphs (Xu et al., 2021), documents (Paranjape et al., 2022), pre-processed vectors (Verga et al., 2021), other PTLMs (Shwartz et al., 2020), search engines (Nakano et al., 2021), to Wikipedia documents as used in this work. To augment the PTLMs, common practice includes creating synthesizing datasets (Wu et al., 2021), adding knowledge to the prompts (Wei et al., 2022; Nye et al., 2021), create demonstrations (Brown et al., 2020), and extending feature vectors (Khattab & Zaharia, 2020).

The contribution of **IAPEK** is orthogonal to above work, as a gated framework to reject annotations or retrieval. Since *Thrust* allows test time estimation on the queries, without labels nor gold answers, the method can be extended to any of these aforementioned settings.

**Hardness and Confidence Estimation in PTLMs.** Much previous work studies the estimation of dataset hardness and model confidence under the context of PTLMs. For dataset hardness, RDA (Perez et al., 2021) measures the hardness as the cumulative area under the loss curves of cross-fold validation on the test set. Point-wise $\mathcal{V}$-Usable information (Ethayarajh et al., 2022) computes the hardness as entropy difference between the feature-provided case and the blank feature case. Sensitivity Measurement (Hahn et al., 2021) measures the dataset difference by computing the variance of loss of the correct labels on a set of neighbor sentences extracted from generative models with masked original sentences as the inputs. These methods achieve great correlation with the model performance. However, all these methods focus on analyzing the test set performance, thus the test set labels are required and can not be applied when predicting the answers. Another line of work focuses on estimating the expected calibration errors (ECE) for classification (Kong et al., 2020), QA (Jiang et al., 2021), and math (Lin et al., 2022) datasets, as a reflection of model certainty on the correct answers. ECE can be considered as an orthogonal evaluation metric to measure the model's capability of understanding the tasks, compared to common metrics such as accuracy.

Most of the previous work can be considered a posterior analysis of the model capability. In this work, instead, we estimate the pragmatic confidence at the test time to empirically increase the performance with limited budget or bandwidth to acquire knowledge.

## 6 CONCLUSION

In this work, we propose **I**nstance-level **A**daptive **P**ropulsion of *E*xternal **K**nowledge **(IAPEK)** as a solution to propel model performance when the external knowledge is useful but noisy. Accordingly, we propose a simple and effective instance-wise metric, *Thrust*, to perform the adaptive knowledge injection. Extensive experiments show that *Thrust* can improve the performance of utilizing external knowledge under various settings. Understanding the delicate usage of potentially noisy knowledge for PTLMs can further enable the models to conduct inference beyond the limitation of implicit internal knowledge.

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

# A APPENDIX

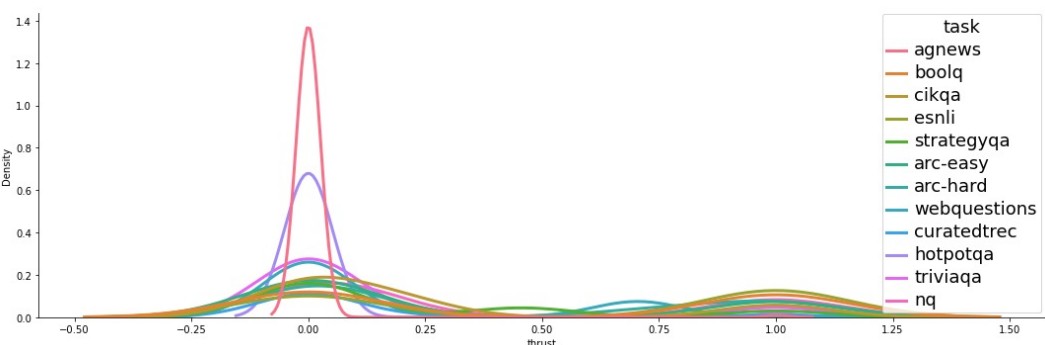

Figure 5: Distribution of **_Thrust_** scores for all involved tasks with UnifiedQA-3b to create the instance representation. The distribution is normalized by Kernel Density Estimation. Low scores denote the cases where internal knowledge is not enough, vice versa.

| Dataset | BM25 25% | BM25 50% | BM25 75% | Thrust 25% | Thrust 50% | Thrust 75% |
|---|---|---|---|---|---|---|
| AGNews | 77.9 \| 77.0 | 80.1 \| 79.2 | 82.3 \| 81.3 | 77.9 \| 78.4 | 80.1 \| 80.4 | 82.3 \| 82.3 |
| e-SNLI | 69.1 \| 68.4 | 75.9 \| 75.6 | 82.8 \| 83.0 | 69.1 \| 86.3 | 75.9 \| 87.5 | 82.8 \| 88.8 |
| CIKQA | 62.7 \| 62.3 | 64.1 \| 64.4 | 65.5 \| 66.1 | 62.7 \| 66.9 | 64.1 \| 66.9 | 65.5 \| 66.9 |
| StrategyQA | 64.1 \| 63.3 | 70.5 \| 68.3 | 77.0 \| 78.1 | 64.1 \| 74.3 | 70.5 \| 81.4 | 77.0 \| 82.9 |
| BoolQ | 68.1 \| 68.4 | 74.6 \| 75.9 | 81.2 \| 82.0 | 68.1 \| 79.1 | 74.6 \| 85.7 | 81.2 \| 87.1 |
| ARC-E | 74.4 \| 74.9 | 75.1 \| 75.3 | 75.8 \| 76.3 | 74.4 \| 74.6 | 75.1 \| 74.9 | 75.8 \| 75.1 |
| ARC-C | 64.5 \| 65.2 | 64.4 \| 66.2 | 64.3 \| 66.6 | 64.5 \| 63.9 | 64.4 \| 64.9 | 64.3 \| 65.6 |
| WQ | 20.9 \| 19.0 | 30.0 \| 28.2 | 39.1 \| 37.3 | 20.9 \| 19.3 | 30.0 \| 35.4 | 39.1 \| 46.4 |
| TREC | 19.6 \| 20.4 | 27.0 \| 28.1 | 34.4 \| 36.3 | 19.6 \| 37.8 | 27.0 \| 40.6 | 34.4 \| 40.9 |
| HotpotQA | 24.9 \| 25.2 | 32.3 \| 32.8 | 39.7 \| 40.6 | 24.9 \| 41.9 | 32.3 \| 43.9 | 39.7 \| 45.7 |
| TriviaQA | 39.2 \| 34.2 | 52.8 \| 50.0 | 66.4 \| 65.4 | 39.2 \| 68.3 | 52.8 \| 71.0 | 66.4 \| 73.4 |
| NQ | 12.8 \| 12.9 | 21.1 \| 21.1 | 29.4 \| 29.6 | 12.8 \| 35.9 | 21.1 \| 36.5 | 29.4 \| 37.0 |

Table 5: Performance of **IAPEK** with UnifiedQA-3b leveraging _Thrust_ and BM25. With 25%, 50%, and 75% percent instances augmented with their corresponding knowledge, performances of random/BM25 and random/**_Thrust_** are presented before/after the vertical bar. If performance increases with **_Thrust_**, the score will be marked in green and otherwise in red. WQ, TREC denote the tasks of Web Questions, Curated TREC, respectively. We can observe **IAPEK** performs good with BM25 as the difficulty score on QA tasks. However, the performance of BM25 is worse (on QA tasks) and less robust (on MC classification tasks) comparing to our **_Thrust_**.

We use BM25 (Trotman et al., 2014), a common approach to evaluate the difficulty of queries, as an alternative to **_Thrust_** to perform **IAPEK**. Specifically, we regard each test input as the query and all training data input as the corpus to extract the score. We use the average of the relevance score across the corpus to rank each test input. From Table 5, we can observe that BM25 leads to performance improvement in many cases, **_Thrust_** shows better (e.g., for QA tasks) and more robust improvement (e.g., for classification tasks).

