# OpenReview forum: "Thrust: Adaptively Propels Large Language Models with External Knowledge"
_ICLR.cc/2023/Conference — Submitted to ICLR 2023_

### Official Review · Reviewer_hhnt · 2022-10-25

**Confidence:** 4
**Correctness:** 2
**Technical Novelty And Significance:** 2
**Empirical Novelty And Significance:** 2
**Recommendation:** 3

**Clarity, Quality, Novelty And Reproducibility:**

- The misalignment between the motivation and the experiments makes the paper’s point unclear.
- Some experimental setup details are missing, making the results difficult to be reproduced for now. For example, the authors did not indicate the number of clusters they used in the experiments, and how many knowledge instances they used to augment a query (which seems to be one).
- In the caption of Table 2, “Performances **with/without** knowledge external knowledge are presented before/after the vertical bar, respectively,” it should be **without/with** based on the description of the results in the paper. Such an important typo makes the paper difficult to understand.

**Strength And Weaknesses:**

**Strengths**

- The proposed method to calculate the Thrust score seems novel.
- It is interesting to use the Thrust score to see if the task requires external knowledge or not.

**Weaknesses**

- The motivation and suggested method are not well-aligned.
- The authors claim that they simulate real-world cases, but the experimental setup is highly artificial, making the practicality of the algorithm dubious.
- The experiments are insufficient to show the effectiveness of Thrust.

**Explanations on Weaknesses**

- The authors claim that they try to simulate the real-world case where they have limited bandwidth to retrieve external knowledge from the perspective of cost-efficiency, but the claim is not persuasive because the way they choose the portion of the queries to use external knowledge is through relative comparison between queries after scoring all other queries, and not through absolute scoring on each independent query, e.g., by applying a score threshold. In other words, the suggested method seems impractical because each query would come to the system independently in the real world, not as a batch as in the experiments. Therefore, if cost-effectiveness is a concern, the authors should have proposed a method that works on each query independently to determine whether that query itself requires external knowledge or not.
- Comparing the results in Table 2 and 3, on many of the tasks, using Thrust to partially utilize the external knowledge drops the performance compared to when all queries are answered with external knowledge (e.g., 70.2 → 56.8 (75% of queries) on e-SNLI UnifiedQA-base, 80 → 73.4% (75% of queries) for TriviaQA UnifiedQA-3b). Therefore, the cost-effectiveness of Thrust, if it exists, comes with a sacrifice of accuracy for many cases, creating a trade-off. Meanwhile, it is also unclear whether the method is actually cost-effective because the calculation of the Thrust score would also take up some inference time. The authors should compare the average inference speed between when all queries use the external knowledge as the input, and when part of the queries use the external knowledge while the Thrust score is calculated for all queries.
- The authors only compare Thrust with random baseline, but in order to show the effectiveness of Thrust as a scoring metric, they should compare the algorithm with baselines simpler than Thrust but better than random, e.g., using BM25 scores.
- [Minor point] While the number of external knowledge instances to utilize and the method of utilizing knowledge might vary, the authors seem to test the effectiveness of Thrust with only one combination of them. The authors did not clearly indicate how many external knowledge instances are used for a query, but it seems like one by inferring from the context. Also, they test only the case of using external knowledge as part of the input and did not explore the cross-attention-based method (e.g., FiD).

**Summary Of The Paper:**

In this paper, the authors point out two limitations in using external knowledge with pretrained language models (PLMs): (1) it is time-costly to index and retrieve on large-scale knowledge bases, and (2) retrieved knowledge could be noisy and misleading. With these limitations as the motivation, they propose a scoring metric named Thrust, which uses the weighted sum of the unit vectors pointing from the query vector (numeric vector for binary classification) to the centroids of the clustered vectors of the training instances which serve as the knowledge to retrieve. The weight is calculated as the size of each cluster over the square of the distance between the query and cluster centroids. The method of using external knowledge in this paper is to use the given knowledge instance as a part of the input through concatenation. As the main experiment to show the effectiveness of the suggested method, the authors came up with an experimental setup where only a portion (25%, 50%, 75%) of the queries could be answered with external knowledge, and used the scoring metric to evaluate all the queries and then choose the portion of the queries with the low Thrust scores. The proposed performs better than the method of randomly choosing the queries to use external knowledge.

**Summary Of The Review:**

The authors’ motivation for this work is interesting, but their proposed method and experimental setup do not align well with their motivation. While they claim to simulate real-world cases, the experimental setup is artificial and highly restricted, making it far from real-world cases. The experiments are insufficient to show the effectiveness of the proposed method, making the usefulness of the algorithm questionable.

---

> ### Author Response · Authors · 2022-11-18
> **Response to reviewer hhnt**
>
> Thanks so much for the valuable comments. Here are some explanations of the questions raised.
>
> 1. the setting in the paper and the setting with a fixed threshold are equivalent in the real-world case with continuing incoming queries. You can regard the current 25% threshold score as a threshold computed from a development set. Then, we can expect that this particular threshold of Thrust will reject the use of external knowledge for 75% of future queries.
> On the other hand, the queries are also computed independently. We only use a set of queries to estimate the clusters and for a new incoming query, we compute its Thrust score with respect to these clusters, not any other incoming queries.
> 2. the claim of cost-efficiency denotes, with a particular amount of knowledge sought, how much performance we can get, which leads to the results in Table 3. In the real case, we can not assume that we already have the knowledge for all the queries (e.g., the human explanations for e-SNLI). Seeking external knowledge is a costly process and Thrust is proposed to point out which queries should be augmented first.
> On the other hand, we still compare the performance of Thrust with the full knowledge case in Table 4 and show that, even with less knowledge, Thrust can get improvement in some cases.
> 3. Thanks for the suggestion on using BM25. We use a random baseline since we propose the instance-level adaptive propulsion of external knowledge (IAPEK). Thrust and BM25 can both be considered as one way to implement it. We add the comparison between random, BM25, and Thrust for completeness in Table 5 in the appendix. Results show that our conclusion still holds that Thrust has better and more robust performance than BM25.
>
> Thanks again for the suggestions on clarity. We have addressed them in the updated draft. The data and code will be released and the results can all be reproduced.

---

### Official Review · Reviewer_xVMQ · 2022-10-25

**Confidence:** 4
**Correctness:** 2
**Technical Novelty And Significance:** 3
**Empirical Novelty And Significance:** 2
**Recommendation:** 3

**Clarity, Quality, Novelty And Reproducibility:**

- A main weakness is that the work is not reproducible. It is not clear if the external knowledge used is published or not, and if and how it is possible to obtain it.
- The methods (and experiments) are described informally, or obscurely, where the specific choices made are not motivated or compared to alternatives.
- The related work section comes at a late phase of the reading, and finally makes it clear that the modeling of explicit knowledge as discussed throughout the paper is not at all a common practice ("In this work, we pioneer the study by adding the knowledge in the plain text format.") This leap is hardly discussed or motivated in comparison to the literature.
- While the motivation for identifying `hard cases' is 'limited bandwidth or budget to retrieve external knowledge', efficiency and computation time is not evaluated.
- Also, the choice or selecting some fixed portion of the queries (e.g., 25 percent) for expansion is not motivated. Why not use a threshold over the query difficulty score for example?
- A baseline of selecting random instances rather the based on difficulty may be informative.
- There are some technical inaccuracies, e.g., the definition of the F1 measure is incorrect.
- There are some typos and style issues.

**Strength And Weaknesses:**

Strengths:
- The topic and general ideas are interesting

Weaknesses:
- A main weakness is that the work is not reproducible, with respect to both the methods and datasets.
- Clarity: some important terms are not defined at all or until later in the paper, including the key term of external knowledge (which commonly means something else actually, as in knowledge graphs).
- The experiments miss some important baselines.
- There are some technical inaccuracies, e.g., the definition of the F1 measure is incorrect.

**Summary Of The Paper:**

The paper uses large pre-trained generative language models to perform tasks of text classification or question answering.
As other works, in classification, they feed the text as prompt, and predict the class with the maximal next-token-probability by the model.
The authors argue and show that by adding to the query some relevant background knowledge in textual form (this text, which is obtained from various sources is referred to as `external knowledge'), performance often improves.
And this leads to the following main claimed contribution of the paper. Rather than incorporate external knowledge for any query, the authors propose to assess the difficulty of addressing each query, and only model external knowledge for the hardest queries. To achieve this, they first cluster the training data within the embedding space, and measure the distance of the query representation to these clusters.

The experiments apply to several language models (including T5, GPT-J and a couple more), either in zero-shot learning setting, or when first presenting the models some labeled examples (fine tuning). Several tasks are evaluated, where the source of `external knowledge' differs per task. For example, in some cases, the document title is the query, and the document content is modeled as external knowledge; in other cases the knowledge pertains to human-authored explanations (on the task of detecting textual entailment between two sentences), relevant human-authored facts (question answering), or facts derived from a knowledge graph.
Results are reported with and without external knowledge. In addition, results are reported when extending only the hardest 25/50/75 percent of the queries using the proposed approach for detecting hard queries. The results are overall positive.

**Summary Of The Review:**

The ideas are interesting, but the writeup is not solid. The presentation of ideas and experiments lack comparison with existing works.
Some terms are not well defined or misused. Reproducibility is a main issue.

---

> ### Author Response · Authors · 2022-11-18
> **Response to reviewer xVMQ**
>
> Thanks so much for the valuable comments. Here are some explanations of the questions raised.
>
> 1. For reproducibility, as stated in the footnote of the first page, we will release the code and data. The explanation of what and how external knowledge is used is described in Section 3.2 and we will release the corresponding knowledge pieces for each query.
> 2. We rewrite the formulas in Section 2.2 and kindly invite you to review the updated version.
> 3. We mention pioneer study as we select one from the various ways to use external knowledge to study the effectiveness of IAPEK and there is open potential to apply it to any other method. As our contribution is orthogonal to designing the ways to use external knowledge, we evaluate our method with one common way to use external knowledge as part of the input, following UnifiedQA (Khashabi et al., 2020).
> 4. For efficiency, the running time for Thrust is around 0.001 seconds per query with around 7 clusters. the running time for DPR for each query is around 1.5 seconds (1 million passages with FAISS to accelerate), which suggests that rejecting X% queries for using external knowledge leads to around X% time savings. We also added a discussion about the computation complexity in Section 3.4.
> 5. The selection of the fixed portion denotes the settings that, if we have a new dataset that we want to apply knowledge-augmented models to and we only have limited time/resources to collect the external knowledge (e.g., human annotations for e-SNLI), can we get good performance on using Thrust to select a portion of queries to annotate.
> 6. We did not use a threshold over the query difficulty since the Thrust scores of different datasets can not be compared, as the instance-level scores are calculated from different training examples and different numbers of corresponding clusters. We did show the distribution of the overall Thrust in Figure 3 and Section 4.2, which suggests that Thrust at the dataset level can help identify the necessity of external knowledge for a particular dataset.
> 7. We use the random baseline since we propose the new set of adaptive knowledge propulsion and much previous work requires a test set label to compute the difficulty, which can not be applied to the run time estimation.
> 8. There is no problem with the definition of the F1 score for OpenQA. The formula computing the F1 score between the extracted span and the gold label is commonly used in the previous work of OpenQA, such as UnifiedQA (Khashabi et al., 2020). We specify it as QA-F1 in the updated draft to avoid confusion.

---

### Official Review · Reviewer_sAvG · 2022-10-26

**Confidence:** 3
**Clarity, Quality, Novelty And Reproducibility:** Overall, the work is presented well a…
**Correctness:** 3
**Technical Novelty And Significance:** 2
**Empirical Novelty And Significance:** 2
**Recommendation:** 5

**Strength And Weaknesses:**

Strengths

+ The core hypothesis of the work makes sense to me, especially for larger models.
+ Proposed method is quite simple and works without further fine-tuning or training and offers efficiency gains.
+ Authors show that using thrust is better at selecting instances which require knowledge than just random selection.

Weaknesses / Questions

- Evaluation with recent models like RETRO [1] and Atlas [2] is missing.
- Table 2, if my understanding is correct, with knowledge case is presented after the vertical bar (|). If that's true, please fix the description of the model which states the opposite.
- Table 2, is the with knowledge same as using full knowledge?
- Table 4, when comparing thrust with full knowledge, what are the exact performance numbers? Please mention the evaluation times for both cases as well.

Citations:

[1] Borgeaud, S., Mensch, A., Hoffmann, J., Cai, T., Rutherford, E., Millican, K., Driessche, G.V., Lespiau, J., Damoc, B., Clark, A., Casas, D.D., Guy, A., Menick, J., Ring, R., Hennigan, T.W., Huang, S., Maggiore, L., Jones, C., Cassirer, A., Brock, A., Paganini, M., Irving, G., Vinyals, O., Osindero, S., Simonyan, K., Rae, J.W., Elsen, E., & Sifre, L. (2022). Improving language models by retrieving from trillions of tokens. ICML.

[2] Izacard, G., Lewis, P., Lomeli, M., Hosseini, L., Petroni, F., Schick, T., Yu, J.A., Joulin, A., Riedel, S., & Grave, E. (2022). Few-shot Learning with Retrieval Augmented Language Models. ArXiv, abs/2208.03299.

**Summary Of The Paper:**

This work focuses on improving the use of knowledge in knowledge-augmented Pre-Trained Language Models (PTLM). Authors hypothesize that using knowledge for all instances can backfire and lead the model towards incorrect predictions, but simultaneously for some instances, external knowledge is required. To combat this issue, a new metric, Thrust, is proposed to select instances where external knowledge is more important. Authors test this new method for efficiency and performance improvements on several datasets and models.

**Summary Of The Review:**

Authors have proposed a new metric to effectively select cases which require external knowledge. Authors have done a decent job of evaluating the method with current literature, however a few important evaluations are missing. Since this paper revolves around retrieval based LMs, authors should have done more evaluation on using thrust in conjunction with some of the well known retrieval based LMs.
Overall, I think in it's current form this work requires more refinement in terms of more robust evaluation and I would vote for rejecting this paper.

---

> ### Author Response · Authors · 2022-11-18
> **Response to reviewer sAvG**
>
> Thanks so much for the valuable comments. We would like to respond to your concerns as follows.
>
> 1. thanks for mentioning these recent models, we added them to the literature discussion. However, we may not include them in the experiments at this stage since our contribution is orthogonal to these models as we focus on evaluating the queries on whether the models require external knowledge instead of how to improve models with retrieval. We also added BM25 as an alternative way to perform IAPEK and the experiments are added in the Appendix.
> 2. thanks for the comments, and we have corrected the problem in Table 2. The “with knowledge” case is the same as using full knowledge.
> 3. The performance of the full knowledge model can be found in Table 2. Due to the limited space, we updated Table 4 in the following way: if the performance difference between full knowledge and using Thrust cases is less than 1%, we marked the entry with *.
> 4. for efficiency,  the running time for Thrust is around 0.001 seconds per query with around 7 clusters. the running time for DPR for each query is around 1.5 seconds (1 million passages with FAISS to accelerate). We also added a discussion about the computation complexity in Section 3.4.

---

### Official Review · Reviewer_xs9i · 2022-10-26

**Confidence:** 4
**Clarity, Quality, Novelty And Reproducibility:** This paper is very unclear.
**Correctness:** 1
**Technical Novelty And Significance:** 3
**Empirical Novelty And Significance:** 2
**Recommendation:** 3

**Strength And Weaknesses:**

Overall the idea of this work is novel, but the description is very hard to follow.
The lack of formal definitions, make it hard to  understand how thrust is computed
e.g., "casting a set of instances ( the training data ) into the representation space "
The writing is mostly sloppy, e.g., c_0 is used before it is defined

The experiment result also need more explanations.
For example in Table 3 it is good to see that using knowledge 25% of the time is as good as using it 75% of the time for many tasks. However, one might wonder
why only the result of UnifiedQA is shown?
why not also compare to the case using knowledge 100% of the time?

**Summary Of The Paper:**

This work aims to improve the efficiency and robustness of knowledge augmented LM.
The intuition is for LM to  decide when an external  knowledge source is needed.
A metric Thrust is developed for this decision, based on the relationship between the query embedding and the clusters of instance embeddings.
1) representation learning (the detail of which is not clear)
2)  k-means clustering on training instance embeddings (the training data is not formally defined)
3)  the Thrust score of a query is computed based on its distance to cluster centers and the length of individual instance embeddings. (the intuition is given in Figure 2, but I have hard time connecting that with the given formula)
4) (I assume some procedure "adaptively" filter queries by their Thrust score, but I cannot find the description.)

Experiment is conducted with several LMs (T5, GPT-J, OPT, UnifiedQA) under zero-shot and transfer-learning setttings for QA and classification tasks.

Retrieval is conducted to the top 25%, 50% or 75% queries based on the Thrust score.


**Summary Of The Review:**

This paper is very unclear.

---

> ### Author Response · Authors · 2022-11-18
> **Response to reviewer xs9i**
>
> Thanks so much for the valuable comments. It seems that some important details written in the paper are missed and cause confusion. Here are some explanations for the questions raised.
>
> 1. the training data is defined and described in Section 3.2, with statistics revealed in Table 1. They are the train split from the original data. For clarification, we only use the training data to extract k-means clusters, not for training the CLS or QA models.
> 2. the Thrust score correlates positively with the size of the clusters and negatively with the distance to the centers of the clusters. We present the analysis in Figure 2 and described in the second paragraph of Section 2.2
> 3. The “adaptive” procedure is described formally in Section 2.1, where Thrust is one potential implementation of the scoring function (We have included BM25 as an alternative in the Appendix).
> 4. c_0 is defined as the center of cluster c before the formula (1) in Section 2.2 in the original draft. We improve the clarity of the formula in our updated version.
> 5. As stated at the beginning of Section 3.4, a pre-condition of the effectiveness of Thrust is that external knowledge is useful for the models. As shown in Table 2, zero-shot models can not utilize external knowledge well. Then, according to the precondition, we only use  UnifiedQA. We compared the case using 100% knowledge, and the findings are revealed in Section 4.4 and corresponding Table 4.
>
> We also updated the formula writing in the new draft and kindly invite you to recheck the updated version.

---

> > ### Comment · Reviewer_xs9i · 2022-12-03
> > **Thanks for the updates. Some questions remains.**
> >
> >
> > 1. task definition:
> > I am looking for a mathematical definition of the task, instead of verbal description of datasets.
> >
> > 2. Thrust score:
> > I am looking for its intuition and connect to Figure 2.
> >
> > 3. The “adaptive” procedure:
> > Thanks for the clarification
> >
> > 4. c_0:
> > Thanks for the update.
> >
> > 5. other models than UnifiedQA:
> > Thanks for the clarification. why not also compare to the case using knowledge 100% of the time?

---

> > > ### Author Response · Authors · 2022-12-08
> > > **Follow-up response on the comments**
> > >
> > > Thanks so much for the follow-up questions! Here are some further explanations to the comments.
> > >
> > > 1. The mathematical definition of the task (what we add to the conventional retrieval augmented models) is included in Section 2.1, which denotes that our target is to select a subset  to inject external knowledge and get the most overall performance. In the later experiments (Table 3), we follow this objective.
> > > We will add mathematical definitions for the classification tasks and question answer tasks in our next version.
> > >
> > > 2. Figure 2 shows three different cases of the relation between a test query representation and clusters estimated by some validation examples. If it is too far away from any training data or close to different clusters (similar to the cases close to the classifier boundary), we hypothesize that these instances are harder for models, compared to those close to the center of a single cluster. The connection is that, our Thrust score is computing the sum of vectors pointing instance representation to centers of various clusters, which is aimed to capture the hypothesized hard cases.
> > >
> > > 3. We compare the case of using knowledge for 100% time in Section 4.1 and corresponding Table 4 and show that sometimes Thrust can get better results than using full knowledge, with faster computation. For Table 3, we consider the IAPEK (i.e., we are restricted to only add a certain amount of knowledge) since in the real world application, we do not have unlimited resources (time or budget) to extract the knowledge for all the examples, as the queries continue to flow in the systems and some knowledge requiring human annotations is hard to be collected.

---

### Author Response · Authors · 2022-11-18
**General response to reviewers and paper update summary**

We sincerely thank all reviewers for their time and energy. All the comments have been helpful and we truly appreciate it.
We have updated the draft of the work and highlighted the main updates in blue. The updates are summarized as follows:
1. We updated the method description in Section 2.2 with definitions and formulas.
2. We updated the description of efficiency gain compared to retrieval-based methods in terms of computation complexity in Section 3.4.
3. We added BM25 as an alternative way to perform IAPEK and compared its performance with Thrust in Appendix.
4. We added the mentioned citations and various improvements on writing.

---

> ### Comment · Area_Chair_aEWX · 2022-11-28
> **Please look at the updated paper**
>
> Dear reviewers,
>
> Could you please look at the updated draft uploaded by the authors and see if you want to change your scores?
>
> Thanks!

---

### Decision · Program_Chairs · 2023-01-20

**Decision:**

Reject

**Justification For Why Not Higher Score:**

Even though the idea is interesting there are several open experimental and technical questions as posed by reviewers that can make the paper significantly better.

(1) Do we need to introduce another subtask or metric when more and more things are moving towards an end to end model? For example, it would be a reasonable experiment to try having a smaller fine-tuned model on top of the LLM and the retrieved result to see if the external knowledge is actually needed for this task or not. (As opposed to identifying it pre-retrieval)

(2) I agree with another reviewer that the method is actually not really applicable to the real world. If you are performing clustering on the set of queries than you always need a pre-available set of data that this need to happen on. In the real world, you will obtain 1 query at a time and hence it will be hard to obtain and then later maintain this cluster. The method needs to be re-run to maintain freshness of the Thrust model.

**Justification For Why Not Lower Score:**

n/a

**Metareview: Summary, Strengths And Weaknesses:**

The paper presents an interesting problem of identifying when a given query needs external knowledge to be answered while assuming that the baseline model in an LLM that is being used to answer the query. The motivation behind the idea is twofold: (1) If external knowledge is not required (the LLM already has this information encoded), then presenting external knowledge leads to degradation of quality; (2) There is also a waste of resources to query an external KB in order to answer the question.

To evaluate whether or not to use external knowledge the authors present a new metric/method called Thrust. The experiments show that there is performance improvement when using this method over using external KB by default.

**Summary Of Ac-Reviewer Meeting:**

n/a